# Advances in Aptamers-Based Applications in Breast Cancer: Drug Delivery, Therapeutics, and Diagnostics

**DOI:** 10.3390/ijms232214475

**Published:** 2022-11-21

**Authors:** Tooba Gholikhani, Shalen Kumar, Hadi Valizadeh, Somayeh Mahdinloo, Khosro Adibkia, Parvin Zakeri-Milani, Mohammad Barzegar-Jalali, Balam Jimenez

**Affiliations:** 1Student Research Committee, Faculty of Pharmacy, Tabriz University of Medical Sciences, Tabriz 5166-15731, Iran; 2NanoRa Pharmaceuticals Ltd., Tabriz 5166-15731, Iran; 3IQ Science Limited, Wellington 5010, New Zealand; 4Drug Applied Research Centre, Faculty of Pharmacy, Tabriz University of Medical Sciences, Tabriz 5166-15731, Iran; 5Research Center for Pharmaceutical Nanotechnology, Faculty of Pharmacy, Tabriz University of Medical Sciences, Tabriz 5166-15731, Iran; 6Liver and Gastrointestinal Diseases Research Center, Faculty of Pharmacy, Tabriz University of Medical Sciences, Tabriz 5166-15731, Iran; 7Pharmaceutical Analysis Research Center, Faculty of Pharmacy, Tabriz University of Medical Sciences, Tabriz 5166-15731, Iran; 8School of Biological Sciences, Victoria University of Wellington, Wellington 6012, New Zealand

**Keywords:** aptamer, drug delivery, cancer diagnostics, cancer therapy

## Abstract

Aptamers are synthetic single-stranded oligonucleotides (such as RNA and DNA) evolved in vitro using Systematic Evolution of Ligands through Exponential enrichment (SELEX) techniques. Aptamers are evolved to have high affinity and specificity to targets; hence, they have a great potential for use in therapeutics as delivery agents and/or in treatment strategies. Aptamers can be chemically synthesized and modified in a cost-effective manner and are easy to hybridize to a variety of nano-particles and other agents which has paved a way for targeted therapy and diagnostics applications such as in breast tumors. In this review, we systematically explain different aptamer adoption approaches to therapeutic or diagnostic uses when addressing breast tumors. We summarize the current therapeutic techniques to address breast tumors including aptamer-base approaches. We discuss the next aptamer-based therapeutic and diagnostic approaches targeting breast tumors. Finally, we provide a perspective on the future of aptamer-based sensors for breast therapeutics and diagnostics. In this section, the therapeutic applications of aptamers will be discussed for the targeting therapy of breast cancer.

## 1. Introduction

Breast cancer is considered as one of the top five causes of mortality among women worldwide since 1980 [1]. Primary diagnosis and therapy are of pivotal aspects in the prognosis of breast cancer. Recently, constant education on breast cancer symptoms and how to spot breast differences require a medical opinion, and complementarily to the current diagnosis techniques such as mammograms, ultrasounds, MRI scans and biopsies, the survival rate of breast cancer is improved.

The most common ways of diagnosis are based on mammogram (imaging). However, molecular diagnosis techniques such as hormone receptor and protein expression methods (ER, PR, HER2, HERmark™), serum methods (CellSearch^®^, Biomarker translation test), and gene expression profiling methods (Mammaprint™, OncoVue^®^ Test, GeneSearch™ breast lymph node, nipple fluid aspiration, DiaGenic BCtect^®^, Oncotype DX^®^, Theros H/ISM and MGISM) are also used [2]. These techniques has resulted in a 1.8% to 3.4% reduction per year in the breast cancer mortality rate in the last 30 years among U.S woman [3]. However, the current mortality rate is still attributable to a late diagnosis because of the patients’ fear of cancer diagnosis and treatment, competing life priorities, financial problems, embarrassment about having a breast examination, use of traditional methods, lack of information and mammography misinterpretation [4]. Therefore, there is still a need to innovate new strategies to reduce the current mortality rates that are less invasive, fast, do not require woman to expose their breasts and do not require expensive equipment or specialized personnel to interpret the results [5]. Every year 2.26 million new patients are diagnosed with breast cancer and 684,966 deaths are recorded globally due to breast cancer [6]. Woman from rural and remote areas are among the most affected; for example, in rural areas in China, where woman have less access to health care facilities and have a lower income, the mortality of breast cancer has increased in the last 25 years [7]. In 2019, in US approximately 232,000 women were diagnosed with breast cancer and of those, 40,000 died [8].

Breast cancer can be classified according to its gene expression categorized below: 1. Expression of receptor of estrogen (ER), 2. Expression of receptor of progesterone (PR), 3. Expression of human epidermal growth factor receptor-2 (HER2), and 4. Expression of Ki-67 as proliferation index. The tumor subtypes are further classified in Table 1.

The proliferation index Ki-67 is an essential indicator of uncontrolled cellular proliferation in malignancy and critical to make a distinction between ‘Luminal A’ and ‘Luminal B (HER2 negative)’ subtypes [9,10,11,12]. Understanding the distinction between different tumor subtypes, has allowed researchers to implement specific treatments for each subtype specially for luminal A and B subtypes [13].

The current treatment strategies for different breast cancer subtypes are often selected according to the molecular classification of the cancerous cells [14].The treatment is highly dependent on the stage of the cancer at the time of diagnosis. Early stages are treated by directed surgery followed by mild chemo or radiation therapy [15,16]. Late stage diagnosis is treated by mastectomy where the breast tissues are completely removed followed by intensive chemo and radiation therapy. Recent advancements with the discovery of new medicines has opened the pathway for medical treatments and classified under four main themes. Some common medicinal treatments are; chemotherapy drugs (doxorubicin, epirubicin, paclitaxel, docetaxel, 5-fluorouracil or capecitabine, cytoxan, carboplatin), hormone therapy (tamoxifen, toremifene, fulvestrant), targeted therapy for HER2-positive malignancies monoclonal antibodies (trastuzumab, pertuzumab, margetuximab), and antibody–drug conjugates (ado-trastuzumab emtansine, fam-trastuzumab deruxtecan), and kinase inhibitors (lapatinib, neratinib, tucatinib) [17]. Despite the advancements, there is still a need to promote new effective methods for diagnosing and treating breast tumors.

Aptamers offer the potential to be a candidate for improving the therapeutic potential of drugs by becoming a targeted delivery vehicle [18,19]. Aptamers are synthetically designed oligonucleotides that are evolved to bind specific targets of interest. The process of developing an aptamer is known as SELEX (Systematic Evolution of Ligands by EXponential enrichment) [20,21]. SELEX, allows one to reach a high level of affinity and specificity to the target by introducing conditioning pressures that will eliminate low-affinity and non-specific oligonucleotides whilst retaining ones that demonstrate high specificity and affinity for the target of interest [22,23]. Aptamers bind their target molecules by forming non-covalent bonds such as hydrogen bonds, forces of Van der Waals, electrostatic forces and π-stacking interactions [24,25,26,27], forming secondary structures such as hairpins, loops and stems, bulges, G quadruplexes and pseudoknots that help in the binding to the target molecule [28]. In comparison to antibodies, aptamers demonstrate benefits such as increased affinity and specificity, durability, and non-immunogenicity and aptamers can be chemically synthesized at significantly low costs [29,30]. Aptamers can be modified post-development with hybridization and conjugation to a variety of nano-particles and nano-polymers that make them ideal for applications including molecular diagnosis (i.e., polyadenine-modified aptamer as electrochemical biosensor of human breast cancer cells) and the identification of biological targets (i.e. detection of *Staphylococcus aureus* by aptamer based biosensors) [31,32,33]. Aptamers hybridized to quantum dots, silicon nanoparticle carbon and gold nanomaterials can also be used for molecular classification of tumors, bio-imaging and drug delivery [34]. For example, aptamers have been recently used as biomarkers to identify tumor hallmarks and flowing tumor cells (i.e., I-AAP (i-motif-based activatable aptamer probe)), prognosis monitoring (i.e., fluorescent DNA nanodevices), and tumor treatment (i.e., Pegaptanib sodium (Macugen), HeA2_3) [35,36]. Aptamers developed to identify breast tumors and tumor subtypes, and applied in treatment and diagnostics are listed in (Table in Section 4). Based on the PubMed database, the keywords “aptamer use”, “breast cancer”, “diagnosis”, and “therapy” aptamers’ role as diagnostic and therapeutic tools in breast cancer have been rising since 2004 (Figure 1). Aptamer use in cancer research and associated applications is still a young field. From the studies which evaluated therapeutic and diagnostic potential of aptamers on breast tumors [37,38], it was realized that systematic assessment of aptamer adoption methods has not been performed yet. In this review, we investigated multiple SELEX methods, explain the utilization of aptamers in treatment and diagnostic of breast cancer, and present a systematic approach to solve the most common issues on aptamer applications.

SELEX consists multiple iterations of seven steps. (A) The random design and synthesis of single stranded DNA or RNA oligo library (the random region usually consists of 35–120 bases). (B) Incubation of the oligo random pool with the target of interest. (C) Separation and removal of unbound oligos (D). Amplification of target bound oligos with PCR techniques. (E) In the case of DNA, strand separation to generate a new pool of single-stranded oligos for the next iteration of SELEX. The process is repeated between four and 20 cycles until the desired affinity and specificity is achieved. Once aptamer candidates have been selected, the aptamer–target binding is characterized [39,40,41,42]. There are different types of SELEX and each type is implemented according to the properties of the target molecule [43]. The SELEX strategy that is chosen for a specific target depends on several factors such as the type of target molecule (i.e., organic and inorganic compounds, peptides, proteins, and bacterial and mammalian cells), the type of sensor platform where the aptamer is going to be implemented (electrochemical, fluorescent, plasmon resonance, enzymatic amplification, etc.), and the matrix or environment where the aptamer is going to be used (blood, saliva, urine, breast milk, buffers, etc.) [44,45,46,47,48,49,50,51,52].

## 2. Affinity SELEX

Affinity column SELEX is suitable for target molecules such as metal ions, small organic and inorganic molecules, peptides, proteins and various cell types [53,54]. The target molecule is conjugated to a solid phase such as sepharose beads, magnetic particles, nitrocellulose membranes or graphene oxide polymers using functional groups such as amines, carboxyl and thiol groups [55]. The type of chemistry used for the target conjugation to the substrate depends on the substrate and the functional group. Following co-incubation of oligo pool with the affixed target, the oligo demonstrating affinity to the functionalized target is separated and enriched using PCR [56].

Blok et al. [57] used a thrombin affinity column prepared with concanavalin A, to find aptamers specific for thrombin.

Xie et al. [58] adopted carboxylated magnetic beads for HBsAg immobilization by some deoxynucleotide aptamers which can bind with high specificity to the surface antigen of hepatitis B virus (HBV) [59]. In immobilized metal affinity chromatography (IMAC), metal ions such as Zn(II), Cu(II), Ni (II), and Co(II) are immobilized on a support to fractionate and purify proteins in solutions; the IMAC method is used to enrich most of the phosphopeptides from whole cell lysates, and it is reported that titanium dioxide (TiO_2_) beads have an equivalent ability to enrich phosphopeptides [60,61].

## 3. Label-Free SELEX

Advances have led to the development of methods of aptamer selection where no chemical conjugation of the target is needed [62,63,64]. Proteins and cells have a much larger physical size when compared to other small molecules [65]. Therefore, separating target bound aptamers from unbound using molecular sizes, is achievable with fewer steps and less time compared to other SELEX techniques. Selection usually utilizes molecular size based cutoff centrifugal columns [66], dialysis [67], capillary electrophoresis [68,69,70,71], and flow cytometry [72,73,74]. Ultracentrifugation and dialysis separate free aptamers from aptamer–protein complexes based on size differences. Long incubation times, large volume requirements and nonspecific adsorption are limitation of these methods. An aptamer synthesized to specifically bind ochratoxin A (a mycotoxin released by by different aspergillus and penicillium species) was developed with dialysis SELEX. The K_d_ is in the nano molar range and it has been used for the determination of ppb quantities of OTA in naturally contaminated wheat samples [67]. Capillary electrophoresis SELEX is based on the difference between the electrophoretic mobility pattern of bound and unbounded aptamer. Aptamers with high affinity for IgE were selected using this method. In this method, the high selectivity and efficiency of capillary electrophoresis decreased the number of selection rounds to four, shortening the SELEX cycles from the usual 12–20 rounds [68,75]. The aptamers specific for IgE have been used in detection of allergic diseases [76,77]. The binding rate and affinity of fluorescein isothiocyanate (FITC)-ssDNA library and NB4 cells has been defined by flow cytometry. The incubation of FITC-labeled aptamers with live NB4, HL60 and K562 cells showed that the aptamer could recognize the NB4 cells while there was no obvious red fluorescence about HL60 and K562 cells [72].

In recent years, considerable effort has been made to select aptamers capable of binding small molecules and not needing the target functionalization step [78]. Investigation of methods led to the development of graphene oxide (GO)-based SELEX, also known as “GO SELEX”, Gold SELEX, and Capture SELEX [79,80,81]. These methods eliminate the need for chemical conjugation of targets onto a solid matrix [82,83,84]. “GO-SELEX” and Gold-SELEX utilizes the inherent affinity of ssDNA to non-specifically adsorb onto the surface of graphene oxide or gold nanoparticles (AuNP) respectively [85,86,87]. Single-stranded DNA non-specifically adsorbs onto the surface of GO and AuNP. When a target of interest is introduced, the oligos demonstrating affinity dissociate from the surface of the GO or AuNP to associate with the target in solution. Separation of target-bound oligos from those still adhered on the surface is performed by either centrifugation for “GO SELEX” or in case of AuNP–SELEX by aggregation of the particles induced by sodium chloride. PCR amplification is used to enrich the oligos bound to the target before subsequent rounds of SELEX are performed. In contrast to the aforementioned approaches, both GO and AuNP (Figure 2) can also be used as separation aids when the target and oligo pool are co-incubated in solution. Once binding between the oligo and the target is achieved, the oligos with no affinity to the target stay freely in the solution, both and GO or AuNP can be introduced to act as a sieve to remove the freely available oligos leaving the target bound oligo’s in solution [88,89]. The supernatant is used as a template in subsequent PCR enrichment to generate a library for the next selection round.

Capture-SELEX (Figure 3) on the other hand has a certain level of fixation that is necessary to assist with the separation of the target bound oligo from oligos that do not have any affinity [90]. Instead of the target being chemically conjugated to a solid matrix, the oligo library is partially hybridized with a complementary probe fixated at the terminal end onto a solid phase (such as magnetic and sepharose beads) [91]. The design of the complementary probe is the critical step in Capture-SELEX [92,93]. It needs to have sufficient energy to maintain the hybridization and only break the hybridization when the interaction or association with the target molecule is strong for the oligo to form a structure with the target, the hybridization is broken [94]. The oligos change their conformation from the hybridized state and into a specific 3D structure relative to the target molecules physical characteristics [95].

## 4. Cell-SELEX Applications

Cell-SELEX applies to whole living cells as the target for choice, by utilizing either cells communicating a pre-identified target protein or as examined underneath, a particular cell sort, with no earlier information of cell-surface marker proteins [96,97]. In the primary approach, by changing the positive choice on cells altered to overexpress a surface protein of intrigued and counter-selection steps on parental cells, aptamers are created that particularly tie to the chosen target implanted in its natural environment [49,98]. Thus, Cell-SELEX can be more beneficial than Protein-SELEX against the same target but in a filtered shape, an approach that might lead to aptamers that are non-functional in physiological conditions [34,99]. In a few cases, cell-targeting aptamers are connected as stand-alone adversarial agents since they meddled with the work of the protein target [34]. Cell-SELEX is used when an oligo library is selected to bind to a target either expressed on a cell surface as a transmembrane protein or the whole cell itself (such as bacterial and viral cells) [100]. The major advantage of Cell-SELEX is that one can select an aptamer to target the protein in its native form unlike the vector expressed and purified versions available as reagents [101]. Cell-SELEX also offers the ability to screen for unique biomarkers on cell surfaces without any prior knowledge [102,103,104]. Generally, to undertake Cell-SELEX, one needs to carry out a positive selection (cells expressing the target molecule) followed by a negative selection (absence of target molecule) [105] explained in Figure 4.

Modified versions of Cell-SELEX have been reported that allow for the internalization of the oligo (aptamer) [106,107,108,109,110]. This allows for the possibility of developing aptamers that facilitate targeted drug-delivery vehicles [111]. The Cell-SELEX-derived aptamers for use in cancer research are tabulated in Table 2. To precisely define the role of aptamers in both the diagnosis and treatment of breast cancer, the research carried out is summarized. Cell-SELEX can offer assistance with recognizing biomarkers displayed on certain cancer sorts or in recognizing metastatic cancer cells from ”normal” cancer cells, as well as drug-resistant cancer cells from drug-sensitive ones [33]. We moreover displayed data on new targets found by Cell-SELEX that have been approved in vivo. These aptamers are undergoing advanced approval for helpful advancement, counting endeavors to improve their less-than-optimal pharmacokinetic and biodistribution profiles in spite of the fact that this may be simpler using the progressions that have seen FDA-approval [112]. While there are also improvements within the monoclonal counteracting agent field that permit for in vitro selection using antibody libraries, it is not the ease of the SELEX preparation, but the stability and costs included, that will see aptamers used for the identification of unique targets for cancer restorative improvement within the future [113].

## 5. In Vivo-SELEX

In vivo SELEX is a novel adoption method of aptamer selection where an oligo library is injected into vertebrate models (animals particularly) to present organ- and tissue-specific aptamers [44,124]. The oligo library is administrated via an intravenous route and upon euthanization of the host, the oligo library is purified. Mie and colleagues [125] examined RNA aptamers in mice with hepatocellular carcinoma and colon using in vivo-SELEX. The authors isolated aptamer p68; as an RNA helicase-binding aptamer, a target molecule expressed in elevated levels in colorectal cancer. Wang et al. [126,127,128] utilized the approach to select aptamers for non-small cell lung tumor cells in humans and mice. They achieved aptamers with elevated affinity and specificity to mouse tumor tissues and cancer cell line.

## 6. Aptamer Utilization for Breast Cancer

Many reports have listed aptamers for diagnosis and therapeutic applications of various cancer targets (Table 2). In particular, various groups have reported the identification of aptamers against biomarkers for breast cancer (Table 3). Among the certified biomarkers against breast cancer, HER2 is considered as one of the most suitable and pivotal biomarkers and applied either for molecular division or for the targeted therapy of breast tumors in medical treatment. Namazi et al. [129] utilized Cell-SELEX to select an anti-HER2 single-strand DNA aptamer (known as H2) that demonstrated affinity to HER2 with a K_d_ of 270 nM. Qaureshi et al. [130] developed a label-free capacitive apta-sensor using H2 and continued micro-electrodes of the capacitor for HER2 identification in solutions. The highly sensitive apta-sensor was used to identify HER2 in specimen within a reasonable level of 0.2–2 ng/mL. In addition to HER2, other biomarkers of breast tumors were targeted to develop aptamers for detection of breast tumor cells [131,132,133,134,135]. Ahirware et al. [136] applied the HT-SELEX method to develop an ERα-specific DNA aptamer. The aptamer was internalized by breast tumor cells positively expressing Erα. Following internalization, the aptamer localized in the nucleus. The aptamer was utilized to represent mRNA level of ERα in breast tumor cells, and the findings were related to IHC identification of ERα in breast tumor tissues that were either positive or negative for ERα. Lie et al. [137] developed a label-free biosensor for a regulator protein, i.e., nucleolin, which has a modulating role in the stability of *Bcl*-*2* mRNA in tumor cells. The AS1411 aptamer was applied on alarm cantilevers in the microcantilever array. AS1411 interactions with nucleolin in clinical samples stimulated surface stress alterations, causing various flaws between the reference and sensor cantilevers [138,139]. The complex showed increased sensitivity with a LOD of 1.0 nM.

Identification of breast tumors particularly by targeting circulating tumor cells (CTCs) in the serum of patients is pivotal for initial prognosis, diagnosis and monitoring of treatment effects of cancer [140]. Aptamers developed from either SELEX or Cell-SELEX is a potential option for identifying breast tumors [141,142,143,144,145]. Caei et al. [146] improved aptamer-based fluorescence detection by targeting mucin 1 (MUC1) using ssDNA aptamers combined with luminescent terbium (TbIII) for sensitive identification of breast tumor cells. In the presence of breast tumors, the aptamer attaches to MUC1 on the cell surface and the stimulation of Tb^3+^ induced fluorescence dictates the positive signal [147]. The apta-sensor displayed great sensitivity towards breast tumors with a confinement level of detection as low as 65 cells/mL. Moreover, Joe et al. [148] developed an apta-sensor that detected HER2 and MUC1 in of MCF-7 cells reaching 10 cells/mL level of detection sensitivity. Li et al. [149,150] combined aptamers with silver nanoparticles using MUC1 aptamers for the imaging of MCF-7 cells. The system could effectively differentiate breast tumor MCF-7 cells from more metastatic A549 human lung cancer and MDAMB-231 breast tumors.

**Table 3 ijms-23-14475-t003:** Aptamer application in breast cancer.

Target	Sequences	Aptamer	Kd(nM)	Purpose	Ref.
Nucleolin	GGTGGTGGTGGTTGTGGTGGTGGTGG	AS1411	47.3	Diagnostic	[129]
SK-BR-3 cells	TGGATGGGGAGATCCGTTGAGTAAGCGGGCGTGTCTCTCTGCCGCCTTGCTATGGGG	S6	28.2	Therapeutic	[151]
HER2	GGGCCGTCGAACACGAGCATGGTGCGTGGACCTAGGATGACCTGAGTACTGTCC	H2	1.8 ± 0.5	Diagnostic	[152]
Mucin-1	GCAGTTGATCCTTTGGATACCCTGG	Muc1	38.3	Diagnostic	[153]
ERα	ATACCAGCTTATTCAATTCGTTGCATTTAGGTG	ERaptD4	33	Diagnostic	[136]
HER2	AACCGCCCAAATCCCTAAGAGTCTGCACTTGT	HB5	18.9	Therapeutic	[154]
MDA-MB-231	GAATTCAGTCGGACAGCGAAGTAGTTTTCCTT	Xlx-1-A	0.7	Therapeutic	[155]
MCF-10AT1 cells	AGGCGGCAGTGTCAGAGTGAATAGGGGATGTA	KMF2-1a	52	Therapeutic	[156]
EpCAM	CACTACAGAGGTTGCGTCTGTCCCACGTTGTCATGGGGGGTTGGCCTG	SYL3C	20.08	Diagnostic	[157]

## 7. Therapeutic Use of Aptamers

The applicability of aptamers is not only limited to analytical systems but also paves a way for therapeutic uses [124,158]. Aptamers with small molecular weights of ~20,000 Da are easy to penetrate tumors tissues so they have been considered as drugs [159,160]. Currently, there are a number of aptamers that have been examined for the treatment of clinical diseases, while some of them have already received FDA approval for the therapeutic utilization, such as Macugen (Pegaptanib Sodium Injection) for treatment of macular degeneration [139,161,162]. AGRO100 is an aptamer that binds to nucleolin, a protein found intranuclear in all cells, but is uniquely expressed on the surface of tumor cells. Pre-clinical testing demonstrated the inhibitory effect of AGRO100 on nucleolin function and proved its anti-cancer effects against lung, prostate, breast, cervical, and colon cancer, as well as malignant melanoma and leukemia.

In vivo therapeutic application of aptamers is limited due to nuclease decomposition and metabolism of aptamers in physiological conditions [163,164]. To reduce the decomposition and enhance effects of aptamer utility in in vivo therapeutics application, the original aptamers generally require chemical alteration. Modifications such as internucleotide linkage with 3–3′ and 5–5′capping in the terminus with an inverted nucleotide [165,166], 2′-substitutions and phosphodiester linkage replacement with 2-F, 2-NH2, 2-OMe, and sugar rings [167,168], incorporating unnatural nucleotides into the oligonucleotide chain [169], cyclization of nucleic acids by linking 5- and 3-termini [170,171], and dialkyl lipid/PEG/cholesterol modifications at the 5′-End [172,173,174] on the oligo backbone inhibit the nucleases from digesting the oligo aptamers.

### Therapeutic Application of Aptamers in Breast Cancer

A therapeutic agent is something that acts directly on the target. Aptamers are able to modulate the performance of the targeted protein or mRNA, and could influence their physiological roles such as the initiation of apoptosis [175,176,177]. Balae et al. [178,179] adopted glutathione-attaching RNA aptamers to induce the apoptosis of tumor cells in breast cancer. The aptamers gathered reactive oxygen species (ROS), responsible for the regulation of caspases function in breast tumors. The AS1411 aptamer, capable of binding the *Bcl*-*2* mRNA-attaching protein nucleolin, was assessed for the capability to stimulate *Bcl*-*2* gene cytotoxicity and instability in MDA-MB-231 and MCF-7 breast tumors [180,181]. The AS1411 aptamer could suppress the homeostasis of MDA-MB-231 and MCF-7 cells, reducing the life-span of *Bcl*-*2* genes in tumor cells, and hinder nucleolin from attaching to the full element of the AU region in the *Bcl*-*2* gene, eventually triggering apoptotic pathway. Varshney et al. [182] adopted RNA aptamers (hTERTapt8.1, hTERTapt7.7, hTERTapt 9.5, and CR4/CR5) that were specified for the sequence of RNA interactive domain 2 (RID2) of Human telomerase reverse transcriptase (hTERT) [183]. It can specifically and tightly attach to the peptide of hTERT and modulate the performance of the enzyme of telomerase in MCF-7 tumor cells, indicating the RNA aptamers’ potential in the treatment of patients with breast cancer [183].

## 8. Drug Delivery Pathways of Aptamer

Chemotherapy is the key therapeutic strategy utilized for the treatment of the tumor cells of cancers [184]. Some chemotherapeutic medicines can act on normal healthy cells in addition to tumoral cells thus leading to multiple adversary effects. Targeted delivery of chemotherapeutic substances could reduce adverse effects improving the specificity and efficacy of the therapeutic agent [185]. Aptamers’ ability to specifically bind to its target allows for aptamers’ utility as a targeted drug delivery system in breast cancer treatment. Liu et al. [152] developed an aptamer to target HER2 and made a complex with doxorubicin (a drug commonly used in breast cancer treatment). The doxorubicin–aptamer complex was used as a targeted drug delivery system to HER2-positive breast tumor cells. The aptamer–doxorubicin complex successfully delivered the complex to HER2-positive breast tumors, while minimal cytotoxicity was reported for normal cells. Dai et al. [186] improved a MUC1-targeting system of drug delivery using a MUC1–aptamer complex. The DNA tetrahedron (Td) contained the drug inside its DNA structure. The complex of aptamer–Td can adoptively attach and present doxorubicin (Dox) to MUC1 positive breast tumors, resulting in increased cytotoxicity towards MUC1-positive MCF-7 breast tumors versus normal cells, which are negative for the MUC1 marker in vitro (*p* < 0.01). Tao and Wei, et al. reported that they can produce upgraded polydopamine (pD)-reformed nano-substance-aptamer bio-conjugates (Apt-pDDTX/NPs) for the therapy of breast tumor cells with in vivo applications [187]. Both in vivo animal and in vitro cell experiments showed that the Apt-pD-DTX/NPs can improve targeted medicinal delivery, decrease the deleterious aspects of the drugs and enhance the wellbeing of the surviving patients in the treatment of breast tumors [188].

In last decades, aptamer nanomaterial conjugates have been used as medicine delivery agents in breast cancer. Beqa et al. [189] generated a novel mixture of nano-material which consisted of gold nano-substances for photothermal treatment of breast tumors [190]. By using gold nanoparticle-decorated SWCNTs with SKBR-3 breast cancer cell specific S6 aptamers, SK-BR-3 breast tumors were eliminated by applying 10 min of laser radiation at 785 nm with a power of 1.5 w/cm^2^. S6 aptamers interact with cancer cells specifically, then conjugated AuPOP-decorated SWCNT aggregates on the surface of cancer cells so that the application of a laser will kill that particular cell [191]. Similarly, other scientists considered utilizing the benefits of gold nanomaterials due to the ability to absorb extended wavelength of plasmon (700–1000 nanometer) [192]. Cell-SELEX was utilized to develop a new KW16-13 DNA aptamer, which had affinity for metastatic tumors in the breast tissue [141]. Conjugation of AuNPs–aptamer (KW16-13–AuNPs) demonstrated a 71-fold affinity for metastatic tumors of breast tissue compared to KMF2–1a-AuNPs [193,194]. Maik et al. [195] conjugated aptamer AS1411 to gold nanospheres (AuNSs) for targeted therapy of breast cancer. AS1411–AuNSs demonstrated high durability both in serum and solutions, and was simply internalized by target cells in higher doses and lead to an elevated anti-cytotoxic/proliferative effects when compared to either the pure AS1411 aptamer or AuNSs [196]. Moreover, compared to unconjugated AS1411 or GNS linked to control oligonucleotides, the injection of AS1411–AuNSs in vivo remarkably suppressed the growth of xenografted tumors in mice without any toxic side effects, indicating AS1411–AuNSs is a suitable therapeutic candidate for clinical application in breast cancer treatment.

In addition to nanomaterials and chemical drugs, aptamers can deliver small interfering RNA for treatment of gene disorders [197,198,199]. Theil et al. [198] attached an RNA aptamer specific for HER2 to siRNA targeting the anti-apoptotic Bcl-2 gene for utility in HER2 positive tumors. The aptamer–siRNA conjugate was internalized by HER2-positive tumor cells and silenced expressed levels of Bcl-2, improving the sensitivity of HER2-positive breast tumors to chemotherapy. Wang et al. [200] developed an aptamer–siRNA conjugate; survivin exerts a negative feedback of the cell death pathway in cancerous stem cells of DOX-resistant breast cancer cells. Epithelial cell adhesion molecule (EpCAM), which is present at low levels in normal epithelial cells is highly overexpressed (up to 800-fold) in many solid cancers. Therefore, utilizing an active targeting system containing the RNA aptamer specific for EpCAM presented an elevated dose of siRNA to cancerous stem cells that silenced survivin and elevated tumor cell chemosensitivity, thus finally inhibiting the cancerous cell growth, and extending the mice survival carrying xenograft mammalian tumors.

## 9. Conclusions

Aptamer research has shown benefits in the field of research associated with treatment and diagnostics of breast cancer when compared to the traditional use of antibodies such as higher target affinity and specificity, non-immunogenicity, stability, simple modification and conjugation to different nanoparticles for medicine delivery. Over the years, few aptamers have been generated and adopted to have affinity for specific biomarkers of breast tumor cells thus enhancing treatment and diagnostic capability in breast tumors. The advancements in aptamer selection methodologies such as Cell-SELEX have allowed for better design of aptamers capable of binding highly specifically to cell surface biomarkers. In addition, in vivo-SELEX has enabled the adoption of aptamers to effectively internalize and bind specific targets in physiological conditions.

Utilization of specific biomarkers for aptamer-based detection of tissues and cells paved the pathway for aptamers utility as targeted vehicles for therapeutic agents in breast cancer. Aptamers targeting breast cancer biomarkers and cells led to the development of apta-sensors for sensitive, precise identification and diagnosis of breast tumors. Moreover, aptamers could be applied as carriers of drug, or drug delivery vehicles for the treatment of breast tumors. Currently, the development of aptamer-based drug delivery systems generally accommodates nanomaterial-based carriers as the complex can improve the efficiency of both the drug loading and release. In these situations, aptamers are used as target specific detection ligands, while the nanomaterial forms the basis as carriers and thus loaded with chemical drugs for the therapeutic dose. Furthermore, aptamers with overall negative charge are able to associate with and attach to positive charge factors without specificity. Furthermore, due to the aptamers’ short length and small physical structure, aptamers are susceptible to rapid bio clearance. Additionally, the accumulation and nonspecific binding could lead to immune stimulation and polyanionic effects. The aforementioned challenges of aptamers reduce the physiological stability and efficiency of aptamers, which considerably confines their clinical applications. To enhance the durability of aptamers in physiological conditions, chemical alterations are often performed with series including amino, fluoro, O-methyl, etc. Variations of aptamers, such as enantiomers or spiegelmers, are being extensively assessed for medicine utilization as they are resistant to nuclease digestion in in vivo condition. Spiegelmers are achieved by using enantiomeric targets during the selection processes. Additionally, aptamer conjugation with nanomaterials may improve the renal clearance.

## Figures and Tables

**Figure 1 ijms-23-14475-f001:**
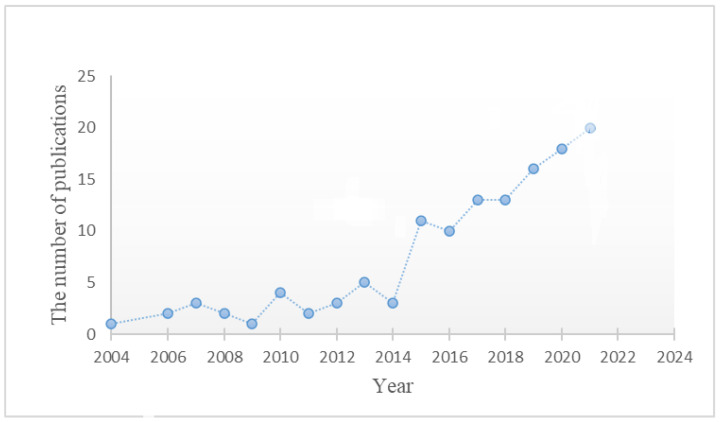
The number of publications on aptamers used as a diagnostic and therapeutic tool for breast cancer since 2004. Keywords included: “aptamer use”, “breast cancer”, “diagnosis”, and “therapy” aptamers’.

**Figure 2 ijms-23-14475-f002:**
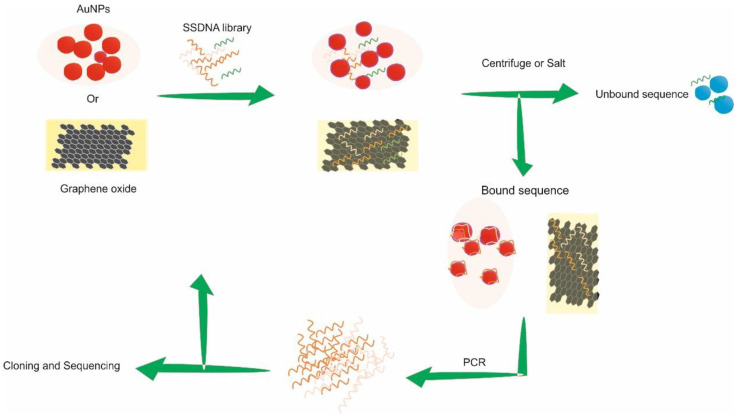
Schematic representation of ‘GO’ and AuNP SELEX.

**Figure 3 ijms-23-14475-f003:**
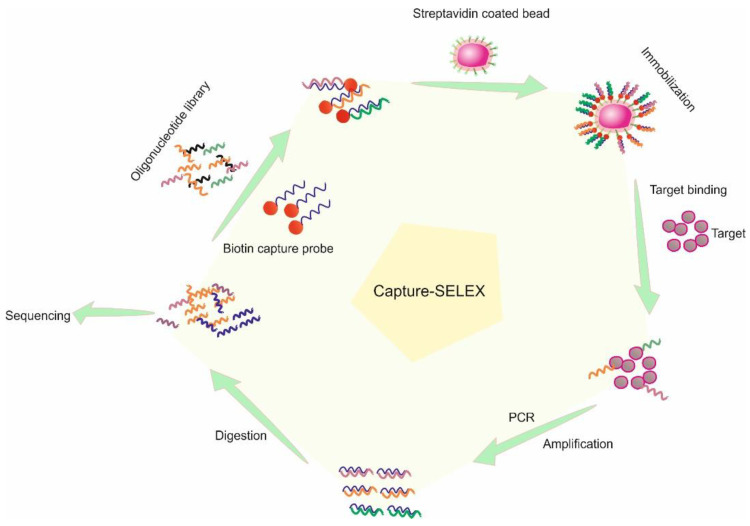
Schematic representation of Capture-SELEX.

**Figure 4 ijms-23-14475-f004:**
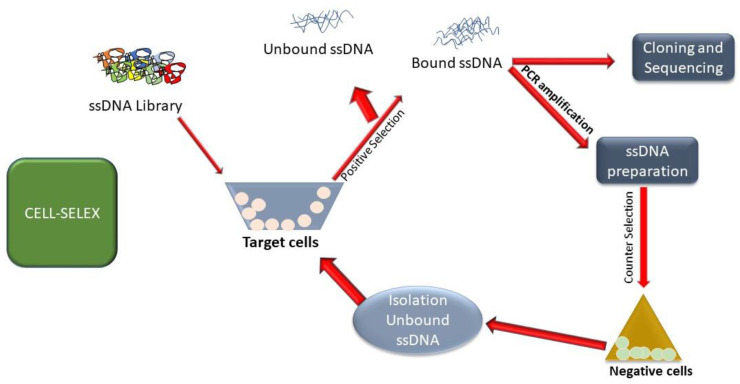
The procedure of Cell-SELEX.

**Table 1 ijms-23-14475-t001:** Tumor subtypes.

Subtype	ER	PR	Ki-67	HER2
Luminal A	+	+/−	<14%	
Luminal B HER2-negative	+	+/−	≥14%	−
Luminal B HER2-positive	+/−	+/−		+
Luminal B Triple negative	−	−		−

**Table 2 ijms-23-14475-t002:** Cell-SELEX-derived aptamers for cancer research.

Target	Aptamer	SELEX Type	Cancer	Purpose	Ref.
MRP1	RNA	Novel combinatorial peptide-cell SELEX	Melanoma	Therapeutic	[114]
CD44/CD24	DNA	Cell-SELEX	Breast cancer	Diagnostic and Therapeutic	[104]
Cytokeratin 19	DNA	Cell-SELEX	Metastatic hepatocellular carcinoma	Diagnostic and Therapeutic	[115]
Alkaline PhosphatasePlacental-Like2 (ALPPL-2)	RNA	Cell-SELEX	Pancreatic cancer	Diagnostic and Therapeutic	[116]
CD133	RNA	Cell-SELEX	Cancer stem cell targeting	Diagnostic and Therapeutic	[117]
EpCAM	RNA	Cell-SELEX	Molecular imaging agents for cancer theranostics	Diagnostic and Therapeutic	[118]
HER2	DNA	Cell-SELEX	HER2 positive breast cancer	Diagnostic and Therapeutic	[119]
PTK7	DNA	Cell-SELEX	Acute lymphoblastic leukemia	Diagnostic and Therapeutic	[120]
Immunoglobin Heavy Mu Chain (IGHM)	DNA	Cell-SELEX	Burkitt lymphoma	Diagnostic and Therapeutic	[121]
AXL	RNA	Cell-SELEX	Human glioma cell cancer	Therapeutic	[122]
CD16-(FcRIII_)	DNA	Cell-SELEX	Cancer immunotherapy	Therapeutic	[123]

## Data Availability

Not applicable.

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
