# Peer review of "Advances in Aptamers-Based Applications in Breast Cancer: Drug Delivery, Therapeutics, and Diagnostics"

_ijms, 2022, doi:10.3390/ijms232214475_

Round 1
Reviewer 1 Report (Previous Reviewer 1)
Thank you for the intresting article! I would like to highlight some points, I will ask the authors to make the necessary changes.
1. "These methods eliminate the need for chemical conjugation of targets onto a solid matrix." - add information about which mechanism of adsorption without conjugation, as well as what kinds of conjugation are commonly used, put the relevant references to the literature.
2. Check the English language
3. "In this review, we investigated multiple SELEX methods, explain the utilization of aptamers in treatment and diagnostic of breast cancer, and present a systematic approach to solve the most common issues on aptamer applications."- there is no information to the SELEX method in the title of the article, although the purpose of the article is to study this method. Change either the aim of the work or the title of the article.
4. Lines 215-224 - there is no literature links...
5. the captions to the pictures should be more informative. You should give more descriptions.
6. Table 2 - there needs to be more discussion in the text about this table. The information that is available does not present any conclusions about the table, that is, it is not shown why it is needed at all.
7. 5. In vivo-SELEX- Give information about the requirements for SELEX for in vivo studies. Explain why so little material has been published on this topic to date.
8. In Figure 1, you show a large increase in publications on the proposed topic, but you only have 6 links to articles from 2020 and 2021 for each year, and only 1 for 2022. It is worth updating your sources of literature.
Author Response
Dear Reviewer,
Thank you so very much for your precious review; all mentioned points have been corrected and highlighted.
Best regards

Reviewer 2 Report (Previous Reviewer 2)
I have no further comments.
Author Response
Dear Reviewer,
Thank you so very much.
Reviewer 3 Report (Previous Reviewer 4)
The authors tried to address the issues raised by the reviewers, however most of them are minor changes and importantly this review article does not have distinct features compared to other ones (Ref 1-3) as I pointed out in the previous reviewing process. I prefer to read other review articles for aptamer-based approaches for breast cancers. Thus, I do not think that this review article meets the high standard in international journal of molecular sciences, and I do not suggest the acceptance of this manuscript.
Ref 1) Cancers 2021, 13(16), 3984
Ref 2) Journal of Nanobiotechnology, 2017, 15(1), 81
Ref 3) Journal of Materials Chemistry B, 2016, 4, 7766-7778.
(1) The newly added sentences (especially, the last sentences in the conclusion section) does not seem appropriate as the last ones in the conclusion section. The authors need to update the conclusion section.
(2) It is desirable to include some figures that represent this review article, which might help the readers to understand the contents.
Author Response
Dear Reviewer,
Thank you so very much.
It is a little disheartening to hear that you would not prefer to read our article, yet we would like to thank you for the time you spent reviewing our manuscript.
Best regards,
Reviewer 4 Report (New Reviewer)
In the submitted manuscript Gholikhani et al. described a potential of aptamer based sensors for breast therapeutics and diagnostics as well as the therapeutic applications of aptamers for targeting therapy of breast cancer. The authors gave readers some perspective remarks related to the future research work.
The article is very interesting and provides up-to-date information, which is confirmed by the latest cited references.
The reviewer suggests that this manuscript should be accepted for publication after minor technical improvements such as:
- Page 2, line 61. In the sentence “Breast cancer can be classified according to its to its gene expression”, “To Its” is two times written.
- Page 2, line 110. The sentence “Based on PubMed database…” is written by larger font. Change it.
Author Response
Dear reviewer,
Thank you so very much.
The required technical improvements have been made, please find the file attached.
regards

Reviewer 5 Report (New Reviewer)
Overall, the review manuscript is well-organized and timely report on recent development of aptamer-based applications in breast cancer treatment and diagnostics. The authors should include some additional examples on applications for drug delivery and explain some mechanisms of “how aptamer function for three different purposes, i.e. diagnostics, delivery and therapy?”
I have following minor comments:
Change title to “Advances in Aptamers-based Applications in Breast Cancer: Drug Delivery, Therapeutics, and Diagnostics”
Line 19-20: Replace “single stranded DNA” with ‘DNA’,
Line 46-47: Reduction rate is not clear. It could be reworded to for clarity.
Line 61: “to its” is repeated in same sentence
Author Response
Dear reviewer,
Thank you so very much all the required changes have been made, please find the attached file.
- The title has been changed.
- the points mentioned below have been adressed.
Line 19-20: Replace “single stranded DNA” with ‘DNA’,
Line 46-47: Reduction rate is not clear. It could be reworded to for clarity.
Line 61: “to its” is repeated in same sentence

Round 2
Reviewer 3 Report (Previous Reviewer 4)
The authors addressed the issues raised by the reviewers, and I suggest the acceptance of this manuscript.
Reviewer 5 Report (New Reviewer)
Authors have addressed most of previous comments.
This manuscript is a resubmission of an earlier submission. The following is a list of the peer review reports and author responses from that submission.
Round 1
Reviewer 1 Report
Thanks for the review! A lot of work has been done. However, there are some minor comments.
1. Slight inaccuracies in the text. There is a colon missing in the sentence "Breast cancer can be classified according to its heterogenous malignancy affecting 1. 61 Expression of receptor of estrogen (ER), 2. Expression of receptor of progesterone (PR), 3. 62 Expression of human epidermal growth factor receptor-2 (HER2), and 4. Ki-67 as prolif- 63eration index. The tumor subtypes are further classified in Table 1.
Character overlap in " there is still a a need to".
The words Additionally and Furthermore are repeated very frequently.
2. I'm not sure I need Figure 1. Especially the last dot on the chart doesn't look very meaningful. Maybe here is a scheme of the 7 steps of SELEX? It would be more informative, especially since it is not very clear from the text these 7 steps. By the way, the text should be rewrote a bit in this direction.
3. the title of the chapter "Label-Free SELEX." should be rewritten. I wouldn't go so far as to say that there are no labels in the techniques described.
4. Figure 4 needs a more extensive explanation.
5. We need explanatory conclusions to table 2.
6. There is a need for uniformity in the writing of abbreviations. For example, "In vivo-SELEX" or "In vivo SELEX".
7. It seems to me that in Conclusion, beginning with "Whilst FDA... " the text could be removed and moved to the main article. The Conclusion should not contain reasoning, it should contain conclusions. We should concentrate our attention on this.
Reviewer 2 Report
The manuscript by Gholikhani et al. contributes to the aptamer knowledge by thoroughly reviewing the recent advances in aptamer applications in breast cancer therapeutics and diagnostics. The manuscript is well organized and generally very well written. The reviewed content is accurate and current.
I have some minor suggestions that could improve the manuscript's readability.
1) Some minor grammatical and editing errors, e.g., Line 30 “breast [cancer] therapeutics and diagnostics, line 214 “bacterial cells and viral [particles],” etc.
2) I suggest adding the Kd values of each aptamer to Table 3. This will help readers recognize the general binding affinities at a glance. This is helpful in a review article.
3) I recognize that some aptamer examples mentioned in later sections, such as section 8, were not categorized in Table 3. Please consider expanding Table 3 or making a new table to summarize all mentioned aptamers.
4) It may be helpful to make a new table that summarizes the diagnostics techniques and LOD (if available).
Reviewer 3 Report
The authors systemically reviewed the development of aptamers and their applications in the treatment and diagnosis of breast cancers. The content is easy to comprehend, however, overall, the review is lacking scientific merit and in depth understanding of aptamers and its advantages and the associated innovations of using aptamers in breast cancer theranostics. Therefore, the review paper is not suitable to be published in IJMS.
1. In Figure 1, the authors may need to explain the reason of the big decrease in publication regarding aptamer in Year 2021. Also, the authors should provide the source of dataset and searching keywords they used.
2. When talking about the SELEX method for the aptamers, the authors should focus more on the different SELEX methods of aptamers for breast cancer theranostics instead of simply discussing all published SELEX methods.
3. The section of "Therapeutic used of aptamer" and "Utilization of aptamer in breast cancers" can be combined. And the authors may need to provide more insightful discussions on this part.
4. As discussed in the paper, aptamers can be conjugated to synthetic biomaterials. Also it can be conjugated to biological vesicles like exosomes for targeted delivery. In terms of breast cancer treatment, the authors need to provide more discussions on current advance in bioconjugation, biomaterials, DNA nanotechnology, aptamer-drug conjugates and how the new methods help to treat breast cancers.
5. In the Conclusion or Introduction part, the authors need to provide a scheme to show the roles of aptamers are playing in the breast cancer treatment. Otherwise, the review paper seems to digress from the main subject.
6. In the Conclusion part, the authors may need to focus on the discussion of how to optimize aptamers for treating breast cancers instead of simply talking about the optimization aptamers and overall challenges in aptamer-based drug delivery.
Reviewer 4 Report
In this review article, the authors summarized aptamer-based therapeutic and diagnostic approaches targeting breast cancers. However, this review article does not have distinct features compared to other review articles (ref 1-3) that provide more in-depth systematic summary of aptamer-based approaches for breast cancers. Furthermore, it did not cover the research published to date (especially, after 2020), but it dealt with the one published in 2010s. Overall, this review article does not meet the high standard in international journal of molecular sciences and thus I do not suggest the acceptance of this manuscript.
Ref 1) Cancers 2021, 13(16), 3984
Ref 2) Journal of Nanobiotechnology, 2017, 15(1), 81
Ref 3) Journal of Materials Chemistry B, 2016, 4, 7766-7778.
(1) The authors need to focus more on the aptamer-based approaches for breast cancer with the recent papers.
(2) The section 2-5 & Figure 1-4 were already summarized in other review articles, so it is not desirable to occupy most of this review article.
(3) The authors need to improve the section 6-8 (diagnostic, therapeutic, and drug delivery of aptamers for breast cancer) and describe more details with appropriate figures.